# One-Year Follow-Up of COVID-19 Patients Indicates Substantial Assay-Dependent Differences in the Kinetics of SARS-CoV-2 Antibodies

Alexander E. Egger,[a] Sabina Sahanic,[b] Andreas Gleiss,[c] Franz Ratzinger,[d] Barbara Holzer,[e] Christian Irsara,[a] Nikolaus Binder,[f] Christoph Winkler,[a] Christoph J. Binder,[g] [ID] Wilfried Posch,[h] Lorin Loacker,[a] Boris Hartmann,[e] Markus Anliker,[a] Guenter Weiss,[b] Thomas Sonnweber,[b] Ivan Tancevski,[b] Andrea Griesmacher,[a] Judith Löffler-Ragg,[b] [ID] Gregor Hoermann[a,i]

[a]Central Institute of Medical and Chemical Laboratory Diagnostics (ZIMCL), University Hospital of Innsbruck, Innsbruck, Austria
[b]Department of Internal Medicine II, Medical University of Innsbruck, Innsbruck, Austria
[c]Section for Clinical Biometrics, Center for Medical Statistics, Informatics, and Intelligent Systems, Medical University of Vienna, Vienna, Austria
[d]Ihr Labor, Medical Diagnostic Laboratories, Vienna, Austria
[e]Austrian Agency for Health and Food Safety (AGES), Department for Animal Health, Moedling, Austria
[f]Technoclone Herstellung von Diagnostika und Arzneimitteln GmbH, Vienna, Austria
[g]Department of Laboratory Medicine, Medical University of Vienna, Vienna, Austria
[h]Institute of Hygiene and Medical Microbiology, Medical University of Innsbruck, Innsbruck, Austria
[i]MLL (Munich Leukemia Laboratory), Munich, Germany

**ABSTRACT** Determination of antibody levels against the nucleocapsid (N) and spike (S) proteins of severe acute respiratory syndrome coronavirus 2 (SARS-CoV-2) are used to estimate the humoral immune response after SARS-CoV-2 infection or vaccination. Differences in the design and specification of antibody assays challenge the interpretation of test results, and comparative studies are often limited to single time points per patient. We determined the longitudinal kinetics of antibody levels of 145 unvaccinated coronavirus disease 2019 (COVID-19) patients at four visits over 1 year upon convalescence using 8 commercial SARS-CoV-2 antibody assays (from Abbott, DiaSorin, Roche, Siemens, and Technoclone), as well as a virus neutralization test (VNT). A linear regression model was used to investigate whether antibody results obtained in the first 6 months after disease onset could predict the VNT results at 12 months. Spike protein-specific antibody tests showed good correlation to the VNT at individual time points ($r_s$, 0.74 to 0.92). While longitudinal assay comparison with the Roche Elecsys anti-SARS-CoV-2 S test showed almost constant antibody concentrations over 12 months, the VNT and all other tests indicated a decline in serum antibody levels (median decrease to 14% to 36% of baseline). The antibody level at 3 months was the best predictor of the VNT results at 12 months after disease onset. The current standardization to a WHO calibrator for normalization to binding antibody units (BAU) is not sufficient for the harmonization of SARS-CoV-2 antibody tests. Assay-specific differences in absolute values and trends over time need to be considered when interpreting the course of antibody levels in patients.

**IMPORTANCE** Determination of antibodies against SARS-CoV-2 will play an important role in detecting a sufficient immune response. Although all the manufacturers expressed antibody levels in binding antibody units per milliliter, thus suggesting comparable results, we found discrepant behavior between the eight investigated assays when we followed the antibody levels in a cohort of 145 convalescent patients over 1 year. While one assay yielded constant antibody levels, the others showed decreasing antibody levels to a varying extent. Therefore, the comparability of the assays must be improved regarding the long-term kinetics of antibody levels. This is a prerequisite for establishing reliable antibody level cutoffs for sufficient individual protection against SARS-CoV-2.

Address correspondence to Judith Löffler-Ragg, judith.loeffler@i-med.ac.at, or Gregor Hoermann, gregor.hoermann@mll.com.

The authors declare a conflict of interest. C.J.B.: Board member of Technoclone GmbH N.B.: employee of Technoclone GmbH, supply with Technoclone ELISA.

**KEYWORDS** SARS-CoV-2, antibody kinetics, assay comparison, neutralizing antibodies, predictive modelling

The severe acute respiratory syndrome coronavirus 2 (SARS-CoV-2) pandemic is still omnipresent, and the number of reported cases and deaths worldwide had exceeded 338 million and 5.5 million, respectively, as of January 2022 (1). The virus contains several structural proteins, of which the nucleocapsid (N) and spike (S) proteins are of particular diagnostic interest. The S protein is a homotrimer consisting of three S subunits, each containing an S1 subunit (located outside the virus membrane and bearing the receptor-binding domain [RBD]) and an S2 subunit as an anchor for the protein in the virus membrane. The RBD is crucial for penetration of the host cell via the angiotensin-converting enzyme 2 (ACE2) receptor; accordingly, neutralizing antibodies against the RBD are important for host response to the virus (2). Upon contact of the virus with the immune system, IgA, IgM, and IgG antibodies against domains of the N and S proteins (including those against the RBD) are formed with different kinetics. While IgA, IgM, and IgG are all found at 3 to 4 weeks of infection, IgM and IgA levels vanish over time, and only IgG antibodies persist longer (3, 4).

Immunological assays have been developed by various manufacturers to measure the humoral immune reaction after SARS-CoV-2 infection or vaccination (5–7). The majority of assays have been calibrated against the first international World Health Organization (WHO) standard (pooled plasma from 11 volunteers with a known history of SARS-CoV-2 infection) using an assay-specific proportional factor and are reported in binding antibody units (BAU) per milliliter (8). While these immunological assays can be performed at large scale, virus neutralization tests (VNT) are considered the gold standard for detecting immunologically active, protective antibodies against SARS-CoV-2. Despite recent developments like pseudovirus VNT or surrogate VNT, mimicking the interaction between the cellular ACE2 receptor and the viral RBD domain in conventional enzyme-linked immunosorbent assay (ELISA) settings, VNT have limited availability and/or may require specific biosafety measures, hampering their broad clinical application (9, 10).

Until now, there has been no consensus regarding the required properties of antibody assays (e.g., antibody class, epitopes, affinity of antibodies), resulting in different specifications among the manufacturers. Consequently, comparisons of results from different manufacturers remain challenging, and comparative studies are often limited to single time points per patient while neglecting differences in the antibody kinetics over time (5–7, 11). Therefore, we determined the longitudinal kinetics of antibody levels (against the N and S/RBD proteins) of patients having recovered from SARS-CoV-2 infection over a year, using assays from Abbott (SARS-CoV-2 IgG assay [Abb_N], SARS-CoV-2 IgG II Quant assay [Abb_S]), DiaSorin (SARS-CoV-2 TrimericS IgG assay [Dia_S]), Roche (Elecsys anti-SARS-CoV-2 assay [Roc_N], Elecsys anti-SARS-CoV-2 S assay [Roc_S]), Siemens (SARS-CoV-2 IgG [sCOVG] assay [Sie_S]), and Technoclone (Technozym RBD IgG assay [Tec_S], Technozym N protein [NP] IgG assay [Tec_N]). In addition, a VNT was performed with all samples as a reference method. Additionally, to the best of our knowledge, we present for the first time predictive modeling of 1-year virus neutralization titers based on antibody levels measured up to 6 months after disease onset.

## RESULTS

**Patient cohort and study time points.** A cohort of 145 coronavirus disease 2019 (COVID-19) patients were studied for their antibody levels and virus neutralization titers at four consecutive visits over 1 year upon convalescence (Table 1). Summarizing all time points, 461 samples from unvaccinated individuals were available. No samples had to be excluded due to a possible reinfection between the study visits. Details on patient fluctuations during the visits are provided in the supplemental material (see Fig. S1).

**Equivalency of the antibody tests.** First, we investigated the qualitative equivalency of the serologic tests (Table S4). The agreement among the assays was quite moderate: 339/461 (74%) of all samples gave concordant results (322/461 positive, 17/461 negative) when all assays were applied. The highest comparability between assays was reached at

**TABLE 1** Patient characteristics and study visit data

| Characteristic | Value |
|---|---|
| Total no. of study patients | 145 |
| At V0 | |
| No. of patients | 122 |
| No. of days since onset (median [IQR])[a] | 60 (50–70) |
| At V1 | |
| No. of patients | 138 |
| No. of days since onset (median [IQR]) | 104 (98–114) |
| At V2 | |
| No. of patients | 125 |
| No. of days since onset (median [IQR]) | 193 (185–202) |
| At V3 | |
| No. of patients (no. unvaccinated) | 107 (76) |
| No. of days since onset (median [IQR]) | 386 (375–394) |
| Gender (male, female) | 82, 63 |
| Age, yrs (median [IQR]) | 56.4 (49.6–69.3) |
| No. of participants aged ≥60 yrs (male, female) | 36, 21 |
| Disease severity[b] (no. of patients [%]) | |
| Ambulatory | 36 (24.8) |
| Hospitalized, mild | 76 (52.4) |
| Hospitalized, severe | 33 (22.8) |
| Polymorbid[c] (no. of patients [%]) | 26 (17.9) |

[a]IQR, interquartile range.
[b]Classified according to the WHO guidelines (12).
[c]Polymorbid indicates ≥4 comorbidities (see the supplemental material).

visit 0 (V0; 60 days after disease onset, when positivity rates were 90% to 94%) and declined over time, reaching the lowest comparability 1 year after disease onset, indicating that initially positive antibody results at V0 turned negative over time to a varying extent among the assays. Positivity rates decreased to 80%, 47%, and 38% for the Tec_S, Tec_N, and Abb_N assays, respectively, while the positivity rates for the other assays (including Roc_N) remained at ≥91%. The equivalency of the assays for individual samples was determined by calculating the root mean square differences, and the results are available in the supplemental material (Table S5).

Regarding quantitative comparability, all assays performed comparably when serial dilutions of the WHO standard were measured (Table S6). Within the sample cohort, the Spearman's rank correlation revealed high pairwise correlation among the Abbott, DiaSorin, Siemens, and Technoclone S protein-specific assays ($r_s$, 0.88 to 0.97) but a surprisingly lower correlation for the Roc_S assay ($r_s$, 0.68 to 0.86). As expected due to the different target epitope, the Tec_N assay showed the lowest $r_s$ value (0.46 to 0.81) with the other assays (Fig. S2).

Modified Bland-Altman plots (Fig. 1) showed a satisfying mean ratio (boundaries of acceptance of ±30%) only between Dia_S and Roc_S (Fig. 1e), Abb_S and Sie_S (Fig. 1c), and Abb_S and Tec_S (Fig. 1d). However, a high proportion of the observations lie outside these boundaries, despite standardization to the WHO calibrator for all assays. The ratios between Sie_S and all assays except Roche and between Abb_S and Tec_S ranged from 0.2 to 5.0 for all observations, while the other combinations, especially those involving the Roc_S assay (i.e., Fig. 1b, e, and h), showed the broadest distributions up to a factor of >10 for individual observations. Combinations with Roc_S also showed a time-specific bias: the early samples (V0; Fig. 1, red) exhibited ratios closer to 1.0, but they gradually diverged over time, with the highest bias in the last samples (V3; Fig. 1, blue). The quantitative Tec_N assay exhibited once more the highest discrepancies with the other assays (Fig. S3).

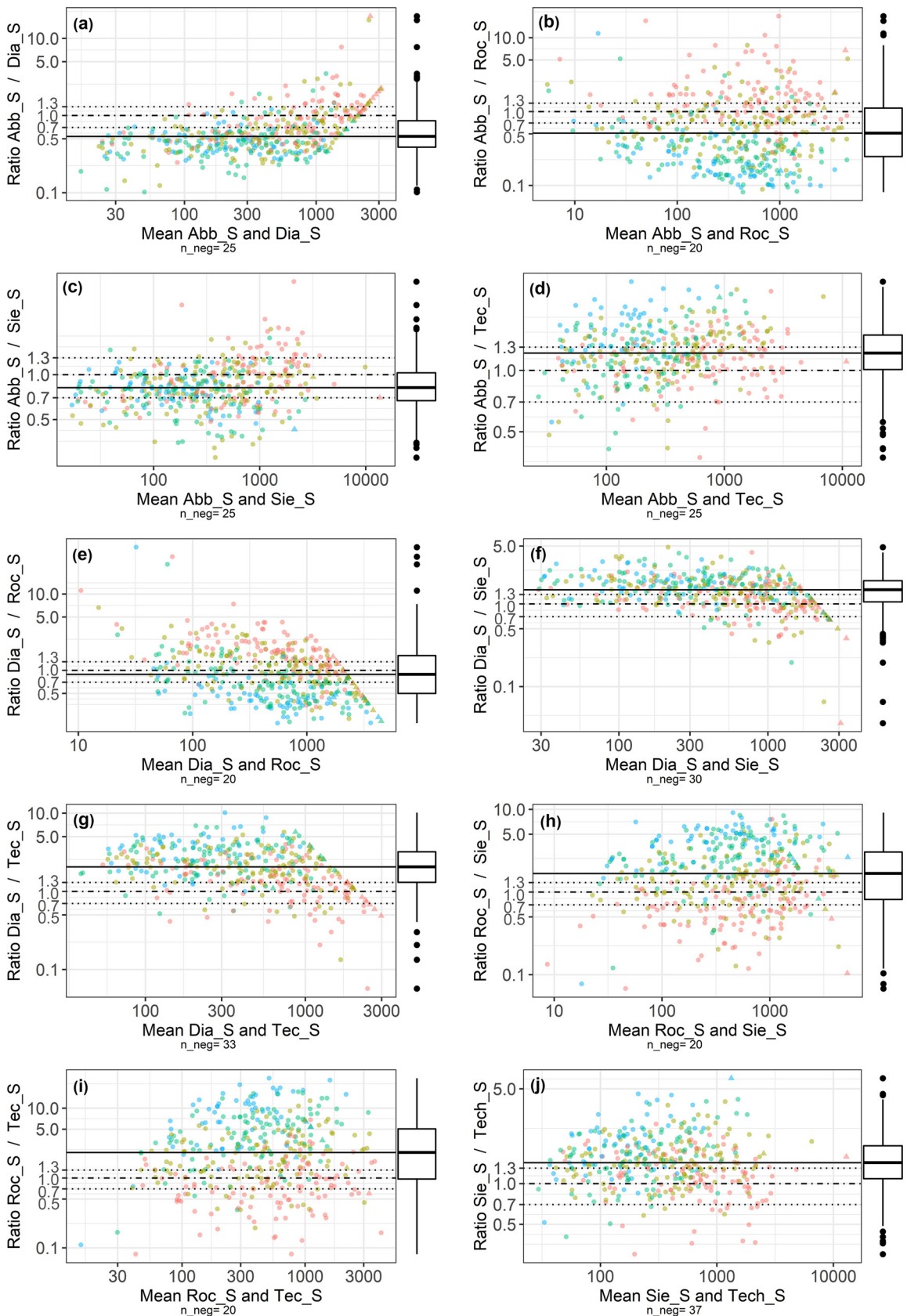

**FIG 1** (a to j) Method comparison plots of all quantitative S protein tests evaluated. In each panel, the *x* axis (logarithmized) indicates the geometric mean of both methods (BAU/mL), and the *y* axis (logarithmized) represents the ratio of the two methods. The solid line indicates

**Correlation with the virus neutralization test.** Next, we compared all tests with the VNT as the reference. Considering all samples available, the qualitative agreement (i.e., concordant positive or negative result) with the VNT was >90% for all S protein-specific assays: Abbott (98.7%), DiaSorin (99.1%), Roche (97.6%), Siemens (98.0%), and Technoclone (92.3%). While Roc_N (98.5%) was comparable with the S protein assays, Abb_N (84.1%) and Tec_N (83.3%) exhibited a lower qualitative agreement with the VNT.

Quantitative comparisons between the methods are depicted in Fig. 2. Increased neutralization titers demonstrated an upward trend to higher antibody levels, but given a specific neutralization titer, the antibody levels showed a broad distribution. Considering all time points together, the highest Spearman correlation with the VNT was calculated for the S protein-specific assays ($r_S$, 0.87 to 0.90), except for Roc_S ($r_S$, 0.77), and—once again—the lowest for the N protein-specific assay ($r_S$, 0.71). Within each visit, the $r_S$ value for Roc_S increased ($r_S$, 0.79 to 0.88) and became comparable to the other S protein-specific assays, while the $r_S$ value for the Tec_N assay remained low ($r_S$, 0.60 to 0.71).

**Time dependence of antibody levels.** Additionally, we compared the serologic assays with respect to the kinetics of the antibody levels (Fig. 3; Table S7). Within 12 months, the virus neutralization titer showed a median decrease to 25% of the V0 level, which was also observed with the Abb_S and Sie_S assays (median decrease to 20% and 23%, respectively). While the median decrease to 36% was slightly less pronounced for Dia_S, both Technoclone assays showed the largest median decrease (to 7% [Tec_N] and 14% [Tec_S] of the V0 level) over time. In strong contrast to these kinetics, the antibody levels measured with the Roc_S assay showed a median increase to 120% of the V0 level up to 6 months and remained nearly constant at this level at 12 months (125% of V0). Details on the distribution of absolute values over time are given in Table S7.

A brief exploratory analysis of subgroups showed that the median antibody levels in males, patients older than 60 years, and multimorbid patients were higher than the levels in females, patients <60 years, and patients not considered multimorbid (Fig. S4). Multimorbid patients showed higher disease severity (Table S8). Antibody levels increased with the degree of initial disease severity of COVID-19. In all subgroups, the Roche assay indicated constant/increasing levels over time, while the other assays indicated decreasing levels (Fig. S4).

**Predictability of long-term virus neutralization titer.** Since virus neutralization titers and antibody levels possess time-dependent kinetics, we questioned whether it would be possible to predict the virus neutralization titer 12 months from the onset of disease based solely on the serological antibody levels measured 6 months after disease onset (or virus neutralization titer up to 6 months as a reference) and clinical baseline variables (age, gender, polymorbidity, immunodeficiency, disease severity).

First, the kinetics of the tests from V0 to V2 were modeled to obtain an average antibody and virus neutralization titer level at 100 days after disease onset (i.e., "intercept 100" in our model) and "slope 30," i.e., the relative change per 30 days. These parameters are shown in Table S9 for each test. The antibody levels measured with the Dia_S, Sie_S, Abb_S, Tec_S, and Tec_N assays resulted in average decreases of 12.9%, 19.8%, 24.1%, 22.5%, and 30.4% per month, respectively, while the Roc_S assay showed an average increase of 6.4%.

Nearly three-quarters of the variability observed in the virus neutralization titer 1 year after infection (V3) is explained by the intercept 100, based on the virus neutralization titer within the first half-year after infection ($R^2$, 0.74; Table 2). Knowledge of only the serologic antibody levels at day 100 leads to slightly lower $R^2$ values for predicting the virus neutralization titer at V3, from 0.65 to 0.69 for the S protein-specific assays and a markedly lower $R^2$ value (0.42) for the N protein-specific assay. Despite the decrease in antibody and virus neutralization titer levels over time (except in the Roche assay results), surprisingly, the slope alone hardly explains variability ($R^2$, 0.00 to 0.39); further, combination of the slope and intercept leads to barely

**FIG 1** Legend (Continued)
median ratio; the dashed line at 1.0 represents the equivalence between both methods; dotted lines show the predefined boundaries of acceptance (±30% of equivalence). Color codes: pink, V0; olive green, V1; green, V2; blue, V3. ▲, observation above the upper limit of quantification for at least one of the two measurements; n_neg, number of observations not displayed due to a negative measurement in at least one assay.

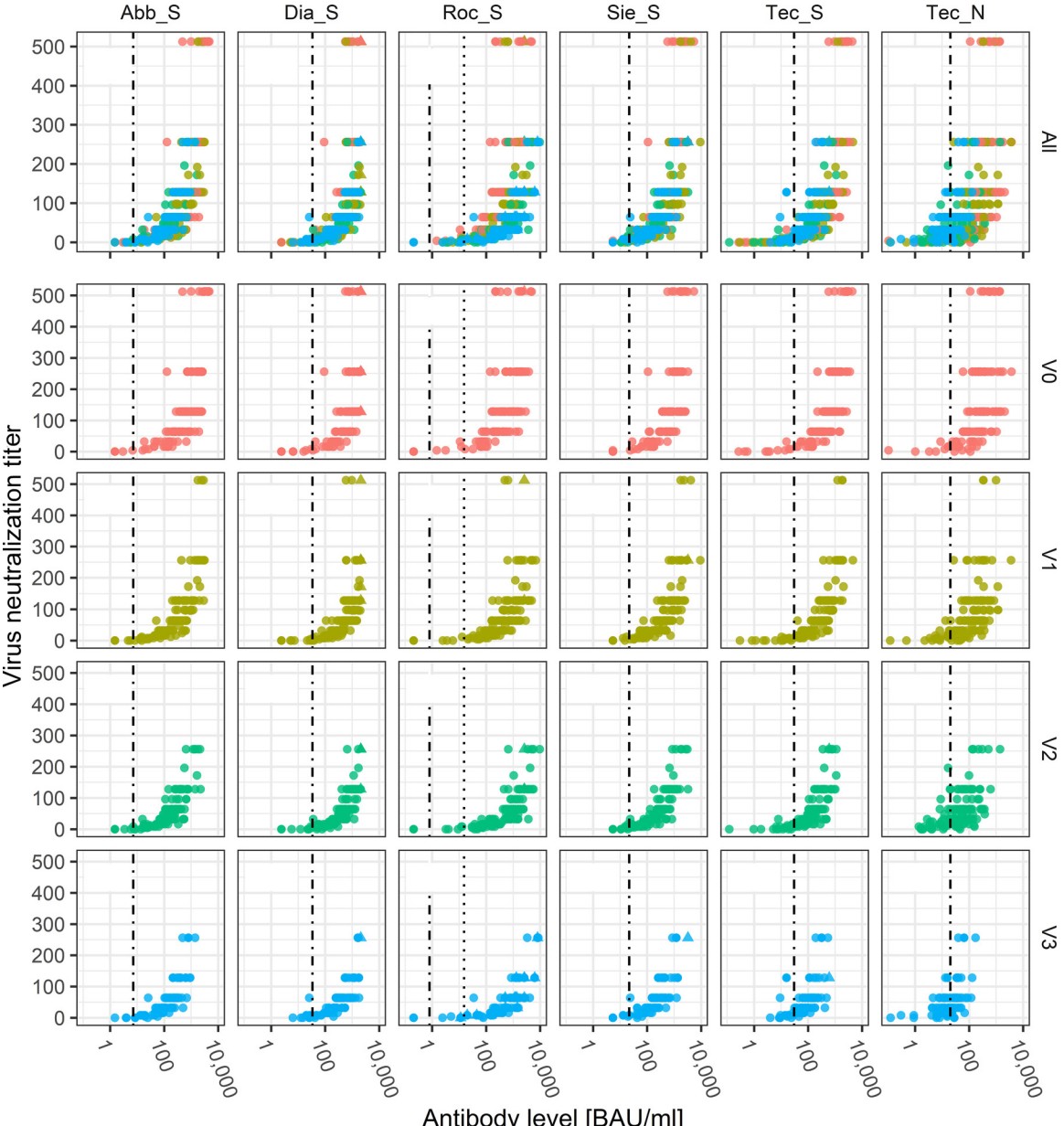

**FIG 2** Comparison of quantitative antibody tests with a live virus neutralization test. Scatterplot of antibody tests (*x* axis, BAU/mL) evaluated against the virus neutralization titer (*y* axis) for all time points and each individual time point (V0, V1, V2, and V3). The Spearman rank correlation coefficient is given in the top left corner of each panel; the dashed line represents the cutoff for positivity—in the case of Roche, a higher cutoff is also stated by the manufacturer for correlation with neutralizing antibodies (dotted line). Pink, V0; olive green, V1; green, V2; blue, V3. ▲, observation above the upper limit of quantification for at least one of the two measurements.

higher $R^2$ values ("$R^2$ both" column in Table 2) compared to those for the intercept alone. The five clinical baseline variables alone give an $R^2$ value of only 0.290 and—once more, unexpectedly—do not produce an added value compared to the sole antibody level at 100 days (cf. the first and last columns of Table 2).

Based on these calculations, we restricted the predictors of the virus neutralization titer 1 year after disease onset (V3) to the serological antibody level 100 days after disease onset for every assay (Fig. 4). The location of the regression line was comparable for all S protein-specific assays (e.g., 100 BAU/mL and 1,000 BAU/mL after 100 days leads to a mean virus neutralization titer of 8 to 16 and 64, respectively, after 1 year). Assays with higher $R^2$ values (Table 2) exhibited narrower prediction limits. However, these limits were still too broad to draw highly confident conclusions for an individual patient: e.g., in the case of Abbott,

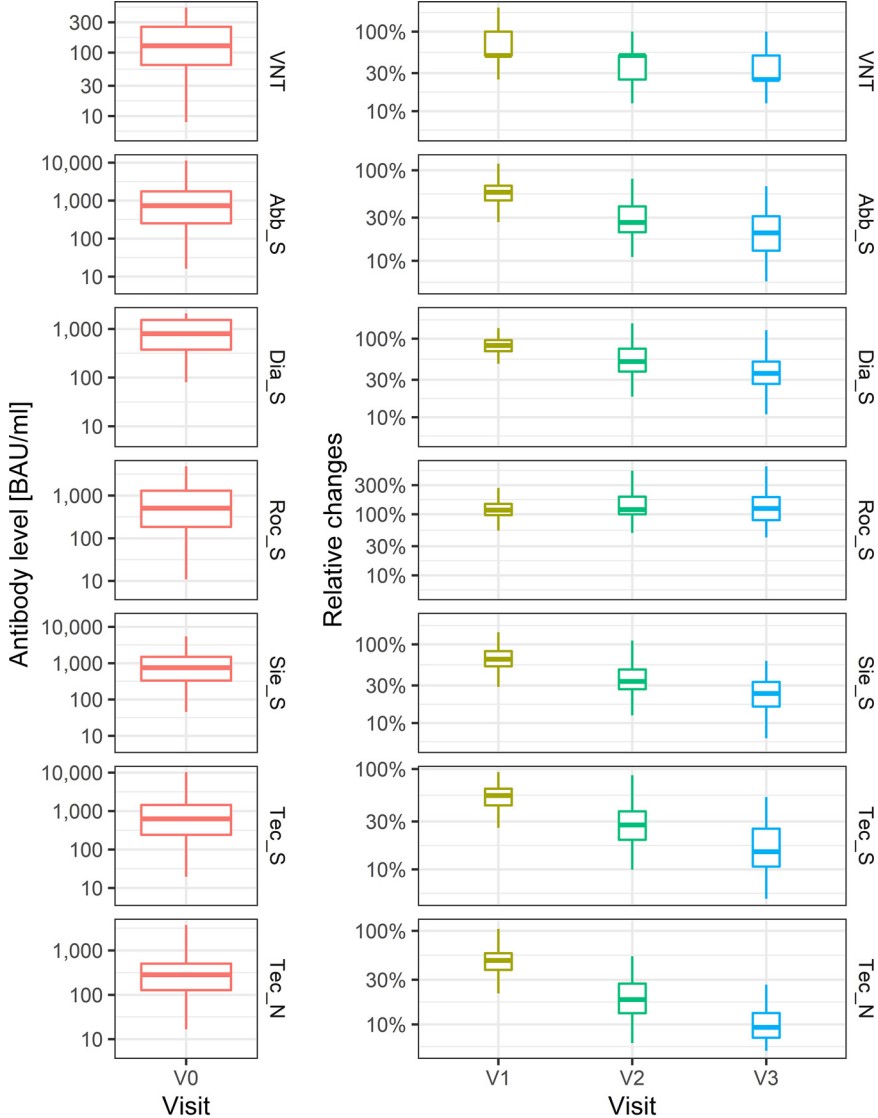

**FIG 3** Boxplots of the absolute (V0) and relative (V1 to V3) values of tests evaluated, showing the kinetics of the antibody levels over time. Pink, V0; olive green, V1; green, V2; blue, V3. Patients without detectable antibodies against SARS-CoV-2 at V0 (*n* = 7) were not considered for the later visits.

100 BAU/mL corresponded (within the 95% predictive limits) to a virus neutralization titer of 2 to 64.

## DISCUSSION

We used a well-characterized longitudinal sample set from COVID-19 convalescents to study the humoral immune response against SARS-CoV-2 for 1 year after infection (12). While anti-SARS-CoV-2 antibodies could still be detected after this period in the majority of individuals, substantial differences between the results of commercial SARS-CoV-2 antibody tests were observed. In particular, the Roc_S test indicated stable results for up to 1 year, whereas all other tests (including the VNT) showed a decline in antibody levels over time. Our results confirmed the Roc_S as a good qualitative surrogate for the humoral immune response after SARS-CoV-2 infection that is largely independent of the time of sampling within the first year after SARS-CoV-2 infection. In contrast, the Roc_S test is not useful for serial antibody measurements in individual COVID-19 patients. In particular, the specific correlation between serologically determined antibody levels and the virus neutralization titer is lost over time.

**TABLE 2** Variables contributing to the variability observed in the virus neutralization test 1 year after infection

| Assay | $R^2$ for:[a] | | | |
|---|---|---|---|---|
| | Int100 | Slope30 | Both | Int100, slope30, and clinical[b] |
| VNT | 0.744 | 0.388 | 0.764 | 0.770 |
| Abb_S | 0.654 | 0.126 | 0.701 | 0.706 |
| Dia_S | 0.662 | 0.005 | 0.689 | 0.694 |
| Roc_S | 0.670 | 0.020 | 0.678 | 0.714 |
| Sie_S | 0.689 | 0.075 | 0.713 | 0.714 |
| Tec_N | 0.422 | 0.003 | 0.433 | 0.494 |
| Tec_S | 0.652 | 0.373 | 0.669 | 0.680 |

[a]$R^2$ values as a measure for the variability explained by the antibody level at 100 days after infection (int100), the slope of the antibody kinetics within 30 days (slope30), and a combination of both ($R^2$ both).
[b]The clinical baseline parameters of age, gender, WHO peak, immunodeficiency, and multimorbidity were additionally considered.

So far, only a small number of studies have investigated the long-term humoral immune response (>6 months) after SARS-CoV-2 infection, comparing only some of the assays we used (Abb_S [13–15], Roc_S [14–16], Abb_N [14, 17–19], Roc_N [14, 16, 19, 20]), or applying other (unspecified) assays (18, 21–26). To the best of our knowledge, Tec_S, Tec_N, Sie_S, and Dia_S were applied for the first time in a longitudinal investigation in our study. In line with our results, a decline in serological antibody levels was reported for all assays except the Roc_S (14–16, 27). The observed differences cannot be explained by the RBD epitope used in the Roche assay, as this is also used in the Abb_S and Tec_S assays. Likewise, measurement of the WHO standard indicated no systematic difference between the assays. The Roche assay is designed to detect high-affinity antibodies against SARS-CoV-2 independent from the immunoglobulin classes, while the other assays investigated in our study are IgG specific. Different dynamics of IgM, IgA, and IgG antibodies are only reported early after infection (3), while the long-term kinetics only exhibited the presence of IgG antibodies (4). In addition, it has been shown that the affinity of IgG antibodies against SARS-CoV-2 increases over time (27–29). In this regard, an increase in antibody affinity could counteract the decline of (other) immunoglobulin levels, leading to stable results for the Roc_S test within the first year of infection (30). In contrast to recently published work, we do not assume that the double antigen sandwich principle (which is only used by the Roche assay) is the primary cause of its detection of high-affinity antibodies (27), as all immunologic tests by Roche share this principle, and in no other immunologic assays from Roche is selectivity toward antibody affinity stated by the manufacturer. While high-affinity antibodies are important in the diagnosis of neurological diseases, their role in COVID-19 has not yet been evaluated sufficiently (31). To overcome the discrepancies observed despite calibration to the WHO standard, further approaches to test harmonization (e.g., nonlinear conversion factors, establishment of a standard based on sera of vaccinated individuals) have been suggested (32). However, a prerequisite for test harmonization in clinical chemistry is an exact definition of the analyte that is still lacking for anti-SARS-CoV-2 antibodies.

A novel aspect of our study is the predictability of the VNT results 12 months after infection, based solely on serological antibody levels 6 months after disease onset. While others have shown that based on several combinations of assays, the disease onset could be calculated retrospectively (33), we herein present the first prospective estimation of the virus neutralization titer based on serologic antibody measurements. We showed that antibody levels 100 days after disease onset explained the majority of the variability of the VNT results after 12 months, while the individual antibody dynamics (indicated by the slope within the first 6 months) and the inclusion of the clinical baseline variables age, gender, polymorbidity, immunodeficiency, and disease severity did not improve predictability further. This could be explained by the fact that many of the investigated clinical baseline variables are known to influence anti-SARS-CoV-2 antibody levels: people younger than 60 or exhibiting a mild disease severity have lower antibody levels, as found in this study and reported previously by

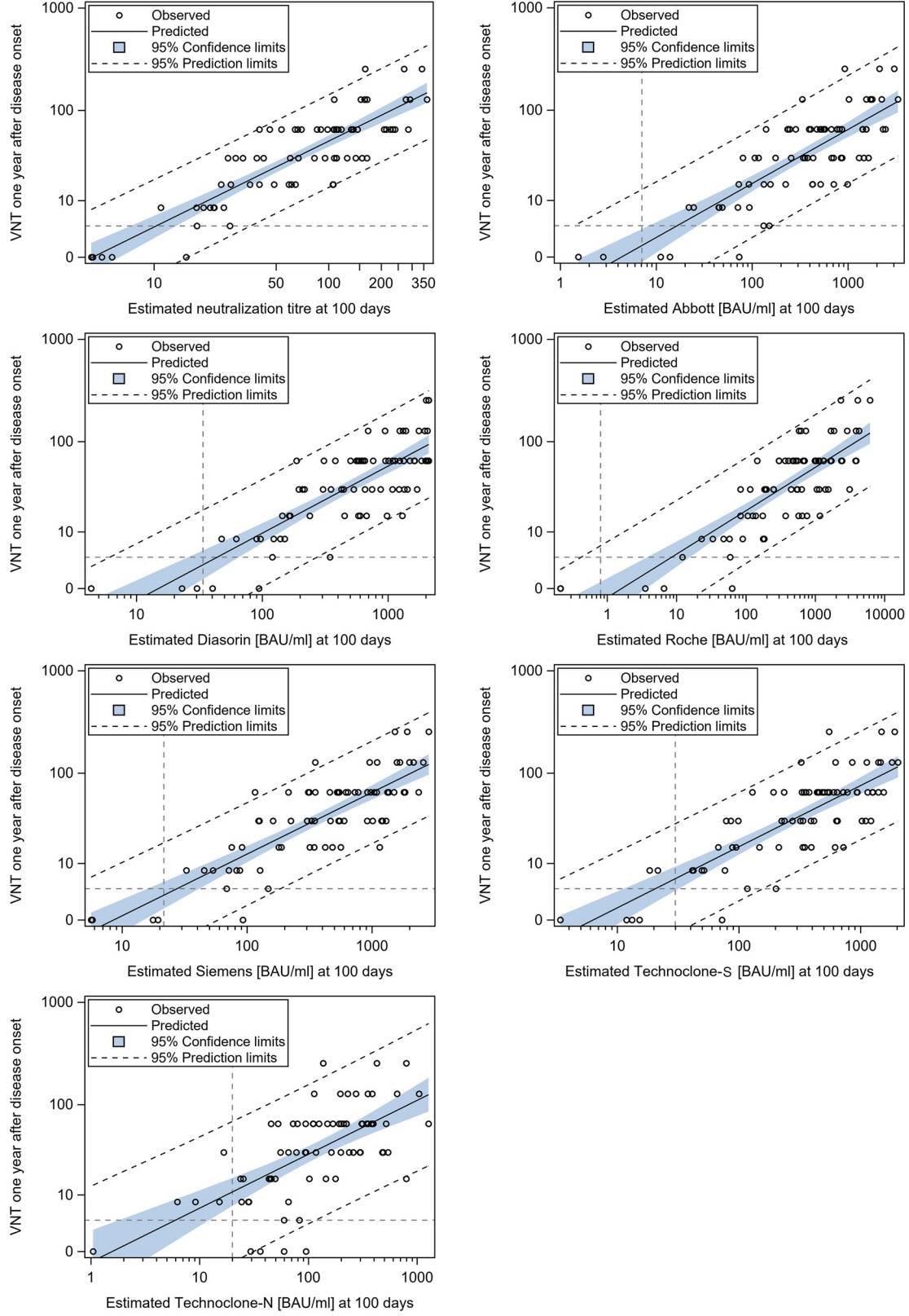

**FIG 4** Predictability of long-term virus neutralization titer. Observed and predicted neutralization titer 12 months after disease onset, based on a single measurement of antibodies or the neutralization titer 100 days after infection. The *y* axis is shown on a shifted logarithmic scale in order to include values of zero. Blue areas, 95% confidence limits for the mean model; dashed oblique lines, 95% prediction limit; dashed horizontal/vertical lines, cutoff for positivity for the respective test; ○, samples (neutralization titer at V3 on the *y* axis; intercept 100 for the respective assay on the *x* axis).

others (34). In our study, we found lower antibody levels for female patients, but sex-specific differences are reported diversely in the literature (34–36). Thus, the effect of clinical baseline variables might be sufficiently incorporated into the antibody level 100 days after disease onset as a summative variable for early immune response. In this regard, a single antibody determination 100 days after disease onset might help to predict the 1-year virus neutralization titer. It has to be clearly emphasized that the sole purpose of our model was to understand the relative importance of antibody dynamics and clinical variables for the long-term humoral immune response against SARS-CoV-2. The association of the antibody level at day 100 and the virus neutralization titer after 1 year still shows wide variability at the patient level. Due to the small sample size and lack of external validation, our model should not be used to predict the long-term virus neutralization titer or even clinical immunity for an individual patient.

While correlations between serological tests and VNT for anti-SARS-CoV-2 antibodies have been investigated in numerous cross-sectional studies (5–7, 37), the strengths of our study are the analyses of a well-defined COVID-19 patient cohort, the serial long-term longitudinal sampling, the direct comparison of four frequently used assays (Abbott, DiaSorin, Roche, Siemens), the inclusion of an independently performed VNT of the same samples, and the mathematical modeling. Regarding the limitations, we included predominately patients with moderate to severe disease, and no asymptomatic SARS-CoV-2-positive individuals were included. It has been shown that disease severity influences the humoral response against SARS-CoV-2 (34). Thus, our results are not necessarily transferrable to asymptomatic or very mild SARS-CoV-2 infections. Also, our cohort was restricted to adult patients, and the results cannot be transferred to a pediatric setting without further studies. Likewise, we studied neither the response to SARS-CoV-2 vaccination nor vaccine breakthrough infections. Again, differences in the dynamics of antibody classes and affinity compared to COVID-19 infections in nonimmunized individuals could influence in particular the relation of the results of the Roc_S assay compared to other quantitative anti-S protein tests. Therefore, our data indicate that a thorough comparison of different serological assays in these situations is necessary before drawing conclusions on the humoral immune response against SARS-CoV-2. Another limitation of our study is that the study patients were infected most likely with the SARS-CoV-2 wild-type virus, which is no longer circulating and which was also used for the VNT. For example, the Delta and Omicron variants of SARS-CoV-2 were found to partly escape the effect of neutralizing antibodies, and the reactivity of the sera tested here might not be sufficient to neutralize these and other emerging SARS-CoV-2 variants (38, 39). Finally, monitoring for reinfections was symptom based. Thus, asymptomatic reinfections can formally not be excluded. However, individual antibody levels did not indicate an immune reaction to a reinfection in the participants.

In summary, we found that among the widely used commercial assays for anti-SARS-CoV-2 antibodies, the Roche (Elecsys anti-SARS-CoV-2 S) assay exhibits substantial differences in longitudinal measurements up to 1 year after acute COVID-19, despite the standardization of results to binding antibody units per milliliter. These limitations of harmonization currently complicate the clinical interpretation of antibody results obtained from the assays of different manufacturers. Once the standardization of antibody levels and definition of protective virus neutralization titers have been accomplished, our model of a prospective 12-month virus neutralization titer based solely on a single measurement may help to estimate the durability of humoral immune response after COVID-19.

## MATERIALS AND METHODS

**Patients and study design.** A total of 145 patients with reverse transcription-PCR (RT-PCR)-confirmed SARS-CoV-2 infection dating from March to April 2020, typical SARS-CoV-2 clinical presentation according to the current WHO guidelines (40, 41), and moderate to severe disease (i.e., either hospitalization or outpatient but with persistent symptoms) were enrolled in the study at the University Hospital of Innsbruck (the main study site) (12). Their antibody status was prospectively evaluated 60 days (visit 0, V0), 100 days (V1), 180 days (V2), and 360 days (V3) after disease onset. Clinical data such as gender, age, severity of acute COVID-19 (42), polymorbidity (≥4 morbidities; see supplemental material), and vaccination against SARS-CoV-2 were obtained for each patient. As vaccine-induced antibodies and antibodies formed due to reinfection would have interfered with our study goal, samples from patients who received their first vaccination >3 days prior

to the study visit or reported symptoms of potential reinfection between the study visits were not considered. All procedures performed in the present study involving human participants were in accordance with the ethical standards of the Institutional and/or National Research Committee and with the 1964 Helsinki declaration and its later amendments. The protocol of the study was approved by the institutional review board of Innsbruck Medical University (EK number 1103/2020), and the trial has been registered at ClinicalTrials.gov (registration number NCT04416100) (12).

**Sample collection and antibody assay investigation.** Blood was drawn via venipuncture and centrifuged (12 min, 2,500 × $g$), and the serum supernatant was stored in aliquots at −40℃. Samples were thawed no more than 3 times to measure S protein-specific antibody levels using quantitative assays from Abbott (SARS-CoV-2 IgG II Quant assay [Abb_S]), Roche (Elecsys anti-SARS-CoV-2 S assay [Roc_S]), Siemens (SARS-CoV-2 IgG [sCOVG] assay [Sie_S]), DiaSorin (SARS-CoV-2 TrimericS IgG assay [Dia_S]), and Technoclone (Technozym anti-SARS-CoV-2 RBD IgG assay [Tec_S]), as well as N protein-specific antibody levels using qualitative assays from Abbott (SARS-CoV-2 IgG assay [Abb_N]) and Roche (Elecsys anti-SARS-CoV-2 assay [Roc_N]) and one quantitative assay from Technoclone (Technozym anti-SARS-CoV-2 N protein [NP] IgG assay [Tec_N]) as a counterpart to the quantitative S protein-based assays. All analyses were conducted according to the manufacturers' procedures. The assays investigated and their figures of merit according to the manufacturer's information are summarized in Table S1 in the supplemental material. In addition, the first WHO reference standard was measured with all assays (8).

Neutralizing antibody titers in human serum and plasma were determined from the same sample by employing a neutralization assay with live SARS-CoV-2 wild-type virus in a 96-well format microtiter plate on Vero 76 clone E6 cells, as described in detail in the supplemental material and by Klausberger et al. (7). A titer of 4 or greater defined a positive result in the assay.

The antibody tests were performed without knowledge of the VNT results and vice versa.

**Data analysis and statistics.** Disease onset was defined as symptom onset; if this information was not available ($n$ = 7/145; 4.8%), the date of the first positive PCR result was used. Categorical variables are described as counts and percentages. Continuous variables are described as median and quartiles, due to asymmetric distributions.

A linear regression model was used to investigate whether VNT or antibody results obtained in the first half-year after disease onset could predict virus neutralization titers 1 year after disease onset. All unvaccinated patients with at least two measurements among the first three visits and availability of VNT at the last visit were included. A random intercept and slope model was used to estimate, for each patient, the result on day 100, which was entered into the linear regression model as an independent variable (see the supplemental material for details). $R^2$ values were used to quantify the proportion of variation in the dependent variable that was explained by the respective (set of) independent variable(s). Estimated regression lines for the assay intercept as the single independent variable are depicted with 95% confidence limits for the mean and 95% prediction limits. Prediction models were calculated using SAS 9.4 (SAS Institute Inc., 2016). Graphs not related to predictive models were plotted using R 4.0.3. (Vienna, Austria) with the ggplot2 library. Further details are available in the supplemental material.

**Data availability.** Anonymized patient data and antibody levels at the respective time points for each assay are available in Data set S1 in the supplemental material.

## SUPPLEMENTAL MATERIAL

Supplemental material is available online only.
**SUPPLEMENTAL FILE 1**, PDF file, 1.6 MB.
**SUPPLEMENTAL FILE 2**, XLSX file, 0.1 MB.

## ACKNOWLEDGMENTS

We acknowledge all the volunteers for participating in the CovILD study. We thank the technical staff at the ZIMCL for faithful and timely analysis of the samples. We also thank Robert Kiechl (Abbott), Harald Schwarz (DiaSorin), Michael Juen (Roche), and Gernot Osterer (Siemens) for valuable discussion and technical support.

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
