## [Reviewer comments · Microbiology Spectrum]

Microbiology Spectrum

One year follow-up of COVID-19 patients indicates substantial assay-dependent differences in the kinetics of SARS-CoV-2 antibodies

Alexander Egger, Sabina Sahanic, Andreas Gleiss, Franz Ratzinger, Barbara Holzer, Christian Irsara, Nikolaus Binder, Christoph Winkler, Christoph Binder, Wilfried Posch, Lorin Loacker, Boris Harmann, Markus Anliker, Guenter Weiss, Thomas Sonnweber, Ivan Tancevski, Andrea Griesmacher, Judith Löffler-Ragg, and Gregor Hoermann

Corresponding Author(s): Gregor Hoermann, MLL Munich Leukemia Laboratory

Review Timeline:

Submission Date:	February 17, 2022
Editorial Decision:	June 2, 2022
Revision Received:	July 10, 2022
Accepted:	September 6, 2022

Editor: William Rawlinson

Reviewer(s): Disclosure of reviewer identity is with reference to reviewer comments included in decision letter(s). The following individuals involved in review of your submission have agreed to reveal their identity: Zin Naing (Reviewer #1)

Transaction Report:

DOI: <https://doi.org/10.1128/spectrum.00597-22>

June 2, 2022

Prof. Gregor Hoermann
MLL Munich Leukemia Laboratory
Munic
Germany

Re: Spectrum00597-22 (One year follow-up of COVID-19 patients indicates substantial assay-dependent differences in the kinetics of SARS-CoV-2 antibodies)

Dear Prof. Gregor Hoermann:

The authors study an interesting area, although data are now available from several different studies of a similar nature. The reviewer has raised important issues around whether reinfection occurred, the differences in antibody persistence between patients with single and multiple morbidities, and data on the type of infecting virus (ie what VOC was likely to be infecting these patients).

I agree these are important issues that should be addressed with a resubmitted manuscript, that would then be considered as modifications of the existing manuscript. The authors are encouraged to address these.

Link Not Available

Sincerely,

William Rawlinson

Journals Department
Reviewer comments:

Reviewer #1 (Comments for the Author):

This longitudinal study investigated level of humoral immune response in COVID-19 patients for up to one year following SARS-CoV-2 infection, thereby evaluating 8 commercially available anti-spike and anti-nucleocapsid antibody assays, as well as virus neutralisation test. I only have a few minor comments.

- (1) The length of study is relatively long, and re-infection is possible in study participants during one year follow-up. How was re-infection being monitored during the study period?
- (2) Live virus neutralisation test was validated using the WHO reference panel and reference standard for anti-SARS-CoV-2 antibodies. Good correlation was observed between neutralising antibody titres determined by virus neutralisation test and WHO assigned International Units. It would be interesting to determine the performance of 8 commercial SARS-CoV-2 antibody assays (in terms of correlation with WHO assigned International Units), using reference panel and reference standards.
- (3) Clinical data including gender, age, severity of acute COVID-19, polymorbidity, etc. were obtained for each participant. Analysis of antibody levels and COVID-19 disease severity in polymorbidity subgroup could not be found.

Staff Comments:

Preparing Revision Guidelines

Please return the manuscript within 60 days; if you cannot complete the modification within this time period, please contact me. If you do not wish to modify the manuscript and prefer to submit it to another journal, please notify me of your decision immediately so that the manuscript may be formally withdrawn from consideration by Microbiology Spectrum.

Reply to Reviewers

Reviewer: 1

This longitudinal study investigated level of humoral immune response in COVID-19 patients for up to one year following SARS-CoV-2 infection, thereby evaluating 8 commercially available anti-spike and anti-nucleocapsid antibody assays, as well as virus neutralisation test. I only have a few minor comments.

- 1. The length of study is relatively long, and re-infection is possible in study participants during one year follow-up. How was re-infection being monitored during the study period?**

Reply: We agree that re-infection could have been an issue during the observation period of one year. Participants have been asked for symptoms associated with a re-infection at each study visit. No patient had to be excluded due to a symptomatic re-infection. Asymptomatic re-infections can formally not be excluded as regular PCR test have not been performed for monitoring within the study. However, individual antibody levels did not indicate an immune reaction to a re-infection. A paragraph on this minor limitation has been added to the discussion section of the revised manuscript.

- 2. Live virus neutralisation test was validated using the WHO reference panel and reference standard for anti-SARS-CoV-2 antibodies. Good correlation was observed between neutralising antibody titres determined by virus neutralisation test and WHO assigned International Units. It would be interesting to determine the performance of 8 commercial SARS-CoV-2 antibody assays (in terms of correlation with WHO assigned International Units), using reference panel and reference standards.**

Reply: We thank the reviewer for raising this point. We measured dilutions of the WHO reference standard with all 6 quantitative assays used in our study. Data are shown in Table S6 of the revised manuscript. In brief, the WHO standard showed a good agreement between the assays. A comment that the observed differences in the long-term follow-up samples could not be explained due to differences in the calibration according to the WHO standard has been added in the revised manuscript.

- 3. Clinical data including gender, age, severity of acute COVID-19, polymorbidity, etc. were obtained for each participant. Analysis of antibody levels and COVID-19 disease severity in polymorbidity subgroup could not be found.**

Reply: We agree that data on multimorbidity had not been analysed in particular. A stratification of antibody levels according to multimorbidity has been added as additional subpanel in Figure S1. The overlap between multimorbidity and COVID-19 disease severity according to WHO has been added as Table S8 in the revised manuscript.

August 29, 2022

Dr. Gregor Hoermann
MLL Munich Leukemia Laboratory
Munic
Germany

Re: Spectrum00597-22R1 (One year follow-up of COVID-19 patients indicates substantial assay-dependent differences in the kinetics of SARS-CoV-2 antibodies)

Dear Dr Hoermann:

The paper is now suitable for publication with responses to the reviewers from the authors.

Your manuscript has been accepted, and I am forwarding it to the ASM Journals Department for publication. You will be notified when your proofs are ready to be viewed.

Thank you for your patience and for submitting your paper to Spectrum.

Sincerely,

William Rawlinson
Editor, Microbiology Spectrum
